# Fruit and Vegetable Purchases in Farmer’s Market Stands: Analysing Survey and Sales Data

**DOI:** 10.3390/ijerph17010088

**Published:** 2019-12-21

**Authors:** Pauline Rebouillat, Sarah Bonin, Yan Kestens, Sarah Chaput, Louis Drouin, Geneviève Mercille

**Affiliations:** 1Institut de Santé Publique d’Épidémiologie et de Développement (ISPED), Université de Bordeaux, 33000 Bordeaux, France; 2Centre de recherche du Centre hospitalier de l’Université de Montréal, 850 St-Denis, Montréal, QC H2X 0A9, Canada; 3Département de médecine sociale et préventive, École de Santé Publique de l’Université de Montréal, 7101 Avenue du Parc, Montréal, QC H3N 1X9, Canada; 4Direction régionale de santé publique, Centre intégré universitaire de santé et services sociaux du Centre-Sud-de-l’Ile-de-Montréal, Montréal, QC H2L 1M3, Canada; 5Département de nutrition, Université de Montréal, 2450 Chemin de la Côte-Sainte-Catherine, Montréal, QC H3T 1A8, Canada; 6Centre de recherche en santé publique, 1301 Sherbrooke Est, Montréal, QC H2L 1M3, Canada

**Keywords:** fruits and vegetables, food access, food environment, farmers’ market, health inequalities, sales data, food purchases, disadvantaged neighbourhoods

## Abstract

Farmers’ market implementation holds promise for increasing access to healthy foods. Although rarely measured, purchase data constitute an intermediate outcome between food environment and actual consumption. In a study conducted with two seasonal Fruits and Vegetables (FV) stands in a disadvantaged area of Montréal (Canada), we analysed how accessibility, perception, and mobility-related factors were associated with FV purchase. This analysis uses a novel measure of FV purchasing practices based on sales data obtained from a mobile application. A 2016 survey collected information on markets’ physical access, perceived access to FV in the neighbourhood, usual FV consumption and purchases. Multivariate models were used to analyse three purchasing practice indicators: number of FV portions, FV variety and expenditures. Average shoppers purchased 12 FV portions of three distinct varieties and spent 5$. Shoppers stopping at the market on their usual travel route spent less (*p* = 0.11), bought fewer portions (*p* = 0.03) and a lesser FV variety (*p* < 0.01). FV stands may complement FV dietary intake. Individuals for whom the market is on their usual travel route might make more frequent visits and, therefore, smaller purchases. The novel data collection method allowed analysis of multiple purchase variables, is precise and easy to apply at unconventional points of sales and could be transposed elsewhere.

## 1. Introduction

Disadvantaged populations are more heavily affected by non-communicable disease in Canada and in other industrialized countries [1,2,3]. They also consume fewer fruits and vegetables (FV) [4,5], which is associated with non-communicable diseases such as obesity [6]. They may also be more dependent on their immediate surroundings and on public transit for food supplies [7,8,9,10,11,12,13], due to limited financial and material resources (e.g., access to a car). Promising strategies to increase access to FV in disadvantaged neighbourhoods with poor access to healthy foods include opening farmers’ markets, or at a smaller scale, FV stands and mobile vendors [14]. Furthermore, those initiatives are promoted by sustainable food system coalitions in major Canadian cities such as Montreal [15], Edmonton [16] and Toronto [17]. Freshness, variety, quality and taste are important attributes of these alternative food supplies that may help FV consumption [18,19,20,21,22,23]. However, evidence on their effectiveness in increasing FV access remains scarce [24,25].

Determinants of individuals’ use of farmers’ markets are similar to those related to other food acquisition practices and include price, location, consumer perceptions of food offer and consumer values [11,12,22,23,26,27,28,29,30,31]. For low-income shoppers, access considerations (e.g., location, accessibility, price and opening hours) stand out as key factors governing farmers’ market use [28,30,32]. Several studies on farmers’ market interventions designed to improve FV accessibility in Canada and elsewhere report sociodemographic determinants of use. While shoppers are primarily middle-aged women [20,21,33,34,35,36,37], associations between educational attainment or ethnicity and shopping patterns are discordant [38]. Market location appears to be linked to customers’ socioeconomic profiles [35,39]. For example, the relocation of an established farmers’ market near a bus terminal led not only to an increase in overall attendance, but also to a diversification of the customer base, attracting more men, youth and individuals from food deserts and low-income neighbourhoods [35]. Upon relocating, the market also experienced a significant increase in the proportion of its customers that use public transit or active transportation.

Most intervention studies use FV intake as a main outcome for evaluating the effectiveness of farmers’ markets [23,24,33,34,40,41,42,43,44,45,46,47,48,49,50]. However, the majority of these studies are based in the United States, where government assistance programs are used as financial incentives for purchasing FV. This limits our ability to differentiate the effects of market use per se from those of the incentive programs, reducing the transferability of findings to other contexts [24]. A few studies conducted either in the USA or in other countries considered a broader population that did not have access to such incentives [34,39,42,51,52,53,54,55]. Those that focus on farmers’ market interventions without financial incentives report an increase in FV intake [34,55], including certain types of FV [42,53] or a perceived increase in intake [34,39,51,52,54]. Study designs include a longitudinal pre–post study [42], a mixed-method cohort study [55], a repeated cross-sectional study [34], two cross-sectional studies [51,52], a case study [39] and a one-group post-intervention study [54].

Although rarely measured, assessing purchases is relevant for evaluating intervention implementation and effectiveness for increasing FV consumption given that purchases constitute an important intermediate variable between the food environment and actual consumption. Collecting purchase data at Unconventional Points of Sales (UPS) such as farmers’ markets and corner stores is a challenge [56] because existing databases using barcodes (e.g., Nielsen) are generally limited to purchases made in supermarkets and other stores equipped with scanning systems [57,58]. Only five identified farmers’ market studies measured food purchases [43,46,56,59,60]. These studies collected data through self-report or by manually recording purchase inventories. However, these two methods impose a heavy burden on both researchers and participants [43,46,56,59,60]. Other farmers’ market studies analyse self-reported use of government incentive programs, government incentive sales data or records of coupon redemption [61,62,63,64,65,66], self-reported amounts spent [67] or total sales [39]. Some studies on convenience stores use purchase data collected using these same methods [68,69,70,71,72,73,74,75]. Given technological advances and changing legal requirements [76] in tracking sales, point of sales (POS) terminals are starting to gain popularity in UPS. They also represent interesting new purchasing data sources. To date, one study has used POS daily sales data aggregated by product category [77].

In the present study conducted among shoppers of two seasonal FV stands in a disadvantaged area of Montréal (Canada) with low FV access, we analysed factors associated with FV purchases, namely, accessibility measurements, individual perceptions of food access and other mobility-related factors. To do so, we developed and tested a novel method for measuring FV purchases based on sales data collected through a mobile POS application. The three derived purchase indicators are the number of FV portions, the variety of FV purchased, and the amount spent per purchase.

## 2. Materials and Methods

### 2.1. Intervention Context and Description

With 1.8 million inhabitants, Montreal ranks as the second-largest city in Canada. The prevalence of poverty and food insecurity are concerning, with 21% of inhabitants living under the low-income threshold [78] and 12.7% of households reported as food insecure as of 2014 [79]. Half the adult population is overweight and one-third has a chronic disease, while only 40% eat the recommended number of daily servings of FV [80]. Differential access to healthy food is an important health equity issue and has been the target of research over the last 15 years [81,82,83]. Although food deserts per se are not common in Montreal [81], it is estimated that one-third of low-income citizens have poor access to fresh FV within walking distance from their home [84]. Public health programs and policy makers increasingly promote farmers’ markets as a viable source of fresh FV, especially in low-income, urban settings [85,86]. The two FV stands assessed in this study were run by a not-for-profit organization, YQQ (Y’a QuelQu’un l’aut bord du mur), and are among the many interventions tackling local population’s physical and economic access to FV funded by the Montreal Public Health Department since 2008.

Both FV stands were located in the Mercier-Hochelaga-Maisonneuve district of Montréal, 1.7 km apart from each other. One was located along a transit route next to a subway station and had two adjacent neighbourhoods, Louis-Riel and Longue-Pointe (Cadillac market). The other was located near a leisure center in Guybourg, a neighbourhood considered as a landlocked food desert (Guybourg market). In these three neighbourhoods, respectively, 16.6%, 20.3% and 20.1% of adults lived under the low-income measure (LIM) [87], meaning they earned less than half of the median household income adjusted for household size [88]. Moreover, respectively, 31%, 23% and 100% of the low-income population had low physical access (<215 ft^2^ FV surface) to fresh FV within walking distance (500 m) from their homes [89].

The Cadillac market first opened on 7 September 2016 and closed its doors on October 28th that same season, for a total of 14 days of operation. Its operating hours were from 13:00 or 14:00 to 18:00 or 19:00. First introduced in 2014 [90], the Guybourg market was also open three days per week, from 7 July to 29 October 2016, from 15:00 to 19:00 on weekdays and from 10:00 to 14:00 on Saturday, resulting in 45 days of operation. With an average of 53 shoppers/day, customer traffic at Cadillac market was much higher than at Guybourg market (16 shoppers/day) [91]. In both FV stands, all purchases were recorded by vendors using the Square Point of Sale application (Square, San Francisco, CA, USA). Users could pay by cash or credit card.

### 2.2. Design and Sampling

A face-to-face on-site survey was conducted among a convenience sample of adult shoppers buying fruits and vegetables, from 21 September to 28 October 2016. Even though Cadillac opened later in the season (Sept–Oct), data at both markets were collected during the same period to control for seasonal variation in FV offer. For both markets, two research assistants were present at all times during the opening hours, except for when the markets were open simultaneously (once a week). In that case, Cadillac market was preferred for data collection due to higher traffic. For both sites, the research assistants interviewed shoppers just after they completed their purchases. Eligibility criteria included being 18 years or older, speaking French or English, having lived in one’s current home since at least 1 July 2016 and not having already completed the survey. If respondents lacked time, the research assistants collected their phone number for a phone interview at their preferred time and noted the time and date of their purchase. Of 326 eligible shoppers approached at Cadillac market, 68% completed the on-site survey (n = 218, including 43 participants by follow-up phone calls). In Guybourg, 101 eligible shoppers were approached, and 65% completed the on-site survey (n = 66, including 6 by phone). All participants were asked to give their verbal consent by the interviewers using a script approved by the Ethics Committee of the Centre Hospitalier de l’Université de Montréal (CHUM) in August 2016 (N.D. 16.128). A more in-depth description of the sampling can be found in Chaput et al. [92].

### 2.3. Measures

Research assistants administered a 38-item questionnaire to shoppers, taking, on average, eight minutes to complete. Questions were extracted or adapted from previous studies [52,93,94,95,96,97,98]. Prior to data collection, the questionnaire was reviewed by the project partners and pre-tested with 9 participants by a research assistant in French and English. A description of the variables can be found in Chaput et al. [92], except for the FV purchase method and indicators that will be fully described below. Briefly, sociodemographic characteristics included age, sex, ethnicity, household income category (before taxes and deduction), household main source of income and education level. FV intake was assessed using the 6-item FV module of the Short Diet Questionnaire, previously validated among a French-speaking population [97]. Perceived access to FV in the neighbourhood was measured by assessing participants’ level of agreement to four statements regarding dimensions of perceived access: availability, physical access and affordability (see Table 1). Participants’ market shopping habits (in markets other than the intervention markets) were reported in times per week or month and used as control variables.

Spatial- and mobility-related variables such as if the intervention market was along the usual travel route (yes/no), the home address and car access (yes/no) were obtained from the questionnaire. Residential exposure to the retail food environment was computed as a relative measure of unhealthy to healthy outlets [99]. Specifically, we calculated the ratio of the sum of densities of unhealthy stores to the sum of densities of healthy stores, from 2010 Enhanced Point-Of-Interest database (DMTI Spatial^®^, ON, Canada) using ArcGIS v10.3 (Esri, CA, USA). For each food outlet category, kernel densities with an adaptive bandwidth of 1% were calculated within a 500 m road network buffer of participants’ homes. The term “healthy” restrictively referred to “outlets that allow for complete meals with fruit and vegetable options”, and included supermarkets, fruit and vegetable stores, natural food stores, and grocery stores. Inversely, “outlets allowing for complete meals but offering few or no fruit and vegetable options” (i.e., convenience stores and fast-food restaurants) were termed “unhealthy” (see Clary et al. article for more details [99]).

Details of participants’ purchases were collected using the POS application Square that was used as a cash register by the vendor. At both markets, sellers entered all purchased products with the corresponding quantity (number of units or weight) directly on the payment platform to calculate the total amount of the transaction. Customers had the option of receiving the invoice by e-mail. The payment platform assigned a unique alphanumeric character string to each transaction that was automatically recorded in an MS Excel format database. We gained access to this database through our partnership with YQQ. The database did not contain any personal or banking information. The first 4 characters of the transaction code (created by the Square application) were used as a unique customer identifier. In order to retrieve respondents’ complete transaction details in the database, the research assistants noted the time and date of the purchase and proceeded to a brief verification of the items purchased. The research assistants used strategies to facilitate the retrieval of the right transaction, such as (1) noting the least popular items rather than the most popular items purchased, and (2) entering the quantity purchased for very popular items (e.g., 6 vs. 12 ears of corn). In the case where a respondent completed two transactions (e.g., forgot to buy something), the details were collected for both transactions by the research assistants, but treated as a unique purchase. 

#### Dependent Variables

Three FV purchase indicators were computed from the database and considered dependent variables of the study: number of FV portions purchased, variety of FV purchased and amount spent. Specific manipulations were needed to calculate the first two indicators, as described below.

Fresh FV were sold by weight or by unit. For FV sold by unit, ten items of each species were weighed by a research assistant in order to establish an average weight for one unit sold at these points of sales. For some vegetables, less than 10 items were available for measurement. In this case, the experimental average was compared with the average weight indicated in the Canadian Nutrient File [100]. When the quantity of a given purchased vegetable was missing for one participant, it was replaced by the average quantity of this vegetable purchased by other participants in the sample.

The percentage of edible portion was determined for each species of FV according to the Canadian Nutrient File. For each species of FV, we calculated the weight corresponding to one serving of FV as recommended in the 2007 Canadian Food Guide.

We computed the number of FV portions for each species of FV purchased according to the formula below, and then summed all items to obtain the shoppers’ total portions of FV in their basket:(1)$$Number\;of\;FV\;portions\;purchased/item\;=\frac{quantity\;\left(grams\right)\times \%\;ofedible\;portion}{\;weight\;of\;one\;portion}$$

The variety of purchased FV was computed according to the number of different fruit and vegetable species purchased, herbs and home-made vegetable preparations (e.g., tomato sauce, relish). Different products were counted as one species if their nutritional compositions were similar; for example, all types of apple were grouped together, but cherry tomatoes were counted separately from all other types of tomatoes, as were red and yellow onions, and green and yellow zucchinis.

The amount spent per transaction in Canadian dollars was extracted directly from the Square database and corrected to remove any non-edible items that may have been purchased (i.e., bag, returnable jar).

### 2.4. Statistical Analysis

Multivariate models were run to analyse factors associated with FV purchases.

All variables in Table 1 were tested for inclusion (except Residing < 1 km of market) using univariate models (*p* < 0.20). All models were adjusted for sociodemographic variables (age, gender, household size, ethnicity, household revenue under the LIM, education). Additional adjustments were made to FV consumption, market shopping habits, number of visits at the intervention market and to the market location. Mobility variables such as car access, home-to-market distance and density ratio were forced into the models. A multivariate linear regression model was used to analyse the number of FV portions purchased. The dependent variable was transformed by taking the square root in order to fit the linear regression assumptions. Analysis of the amount spent was performed using a linear regression model (without transformation). Linearity in the logit of the dependent variables was evaluated using the Box–Tidwell procedure [93]. Given the distribution of the FV variety variable, a Poisson regression model was used. As over-dispersion is a common issue in Poisson regression models, it was checked for using Pearson’s chi-square ratio.

First, the Multiple Imputation by Chained Equations procedure (R package ‘mice’) was used to handle missing values. Missing values were observed mostly for household income (15.2%, *n* = 39), market shopping habits (2.0%, *n* = 5), and perceived affordability of FV in the neighbourhood (3.9%, *n* = 10). A total of 27 observations were excluded because of incoherent and non-imputable geographic information (postal code), leaving 257 participants for these analyses. Five imputed datasets were generated and then pooled. The imputation method was defined according to variable type. All variables considered in each model were included in the multiple imputation procedure. The household main source of income was added as a predictor in the multiple imputation models.

Descriptive and bivariate statistics were conducted using SAS 9.2 statistical software and multiple imputation performed using R (3.5.1) package ‘mice’. The level of significance was set at α = 0.05.

## 3. Results

### 3.1. Descriptive Statistics

Table 1 describes the characteristics of the whole sample. Shoppers were mostly women, born in Canada, between 18 and 44 years old, with completed postsecondary education. Nearly a quarter of the respondents were under the low-income measure, which is slightly higher than in the studied neighbourhoods (16.6% to 20.3%). The main source of income was salary or self-employment.

Regarding spatial and mobility variables, slightly more than half of the shoppers had access to a car and two-thirds resided ≤1 km from their market. The market was located on the usual travel routes of most participants (77.8%).

Forty-two percent of shoppers ate less than 5 FV a day, with an average of 4.6 portions of FV per day. Shoppers’ perceptions of FV affordability in their neighbourhood were split between positive opinions and negative/mixed opinions. In the two neighbourhoods, 34.2% and 30.7% of shoppers reported not having easy access to fresh FV within walking distance from their homes or on their usual travel route. Twenty-two percent were infrequent market shoppers (less than once a month).

On average, shoppers purchased nearly 12 portions of FV and three distinct varieties of FV, and spent 5$ per transaction.

### 3.2. Factors Associated with FV Purchasing Practices

Table 2, Table 3 and Table 4 show results of the multivariate regression models used to analyse purchasing practices in the two markets, respectively, the number of FV portions purchased, the amount spent and the variety of FV purchased.

#### 3.2.1. Sociodemographic Variables

Attending Guybourg market was significantly associated with a higher number of FV portions purchased (β = 0.87, *p* = 0.003), but not with the variety of FV purchased or the amount spent. Shoppers born in Canada bought cheaper baskets ($1.50 CAD less, *p* = 0.04), with fewer FV portions (β = −0.81, *p* = 0.004) and a lesser variety of FV (β = −0.24, *p* = 0.02). A lower variety of FV purchased was also associated with a larger household size (β = −0.10, *p* = 0.02). Other sociodemographic variables such as income or education level were not significantly associated with purchase variables.

#### 3.2.2. Spatial- and Mobility-Related Variables

When the market was located on their usual travel route, shoppers spent one dollar less (*p* = 0.11), bought fewer portions (*p* = 0.03) and a lower variety of FV (β = −0.25, *p* = < 0.01). Having access to a car, distance between the market and home and residential exposure to the retail food environment were not associated with FV purchases.

#### 3.2.3. FV Consumption, Perceived Access and Shopping Habits

FV consumption level was not associated with the number of FV portions purchased nor with the amount spent. Results revealed a weak association between FV consumption level and the variety of FV purchased (0.04; *p* = 0.04). No clear tendency was observed for perceptions of FV affordability or ease of finding good quality FV in the neighbourhood.

The amount spent was on average $2.50 CAD lower for the shoppers who came only once to the intervention market compared to shoppers who came more than once a month. The number of visits was also significantly and negatively associated with FV variety (β = −0.34, *p* = 0.009).

## 4. Discussion

This study conducted in low-income neighbourhoods of Montreal (Canada) is among the first to analyse FV purchases among urban shoppers in FV stands. It also uses an innovative method for measuring purchases that could be applied to other contexts. Unconventional Points of Sales (UPS) are part of the foodscape in Canada and in other countries and it is important to include them in studies addressing the food environment. However, studies examining food purchases are mostly limited to existing databases using barcodes or self-reporting, thus limiting our possibility of a comprehensive understanding of food purchasing practices including UPS. As more and more interventions implementing UPS are developed, it is important to elaborate reliable tools for monitoring, comparison and evaluation.

Both FV stands studied were located in low-income neighbourhoods with low FV access; one along a transit hub (Cadillac) and the other in a landlocked food desert (Guybourg). While results detected no difference between the two FV stands with respect to variety purchased, amount spent, or FV species offered, they showed that shopping at Guybourg market was associated with a higher number of FV portions purchased. One explanation could be that Guybourg shoppers made cost-saving FV choices (eg., more potatoes and squash; data not shown). Another plausible explanation could be that a higher proportion of Guybourg shoppers had access to a car (87.5% vs. 46.8%, cf. Appendix A
Table A1), which facilitated the transportation of larger amounts of food and heavier foods. Guybourg neighbourhood is surrounded by non-residential buildings and has poor public transport service, meaning that car access greatly facilitates travel within and outside the neighbourhood. This explanation would suggest that physical accessibility does affect FV purchasing habits at this particular market by facilitating the purchase of more FV portions.

Most shoppers lived in close proximity to their market (66.2% ≤ 1 km), but a higher share of Cadillac market shoppers lived further away (34 vs. 25%, cf. Appendix A
Table A2) and reported the market to be located on their usual travel route (82.1 vs. 62.5%, data not shown). This underlines the potential for interventions implemented near transportation hubs that ease the combination of activities or errands. In fact, it has been reported previously that shoppers preferentially choose farmers’ markets that add the least amount of time to their travel rather than the farmers’ market that may be closest to their home [101]. Multivariate analyses also reveal a lower number of FV portions and variety of FV purchased when the market was on the travel route (*p* = 0.03 and *p* < 0.01). It is possible that shoppers living closer to the markets attended more frequently and, therefore, purchased smaller volumes. Amounts of FV purchased at the markets were also not associated with overall FV consumption, suggesting that FV stands to act as a complementary source of FV, which are otherwise mostly bought in conventional points of sale such as supermarkets. Longitudinal follow-up of shopping patterns could reveal further information with respect to mobility patterns and the importance of secondary food sources for achieving healthier diets. Temporal shopping patterns were not analysed in this study, as we collected information on only one shopping event per participant.

Although physical distance from home to market was controlled for in the present study, a more detailed analysis of daily mobility patterns in light of the environmental circumstances surrounding the market could provide additional insights. Spatial dimensions are rarely reported in existing FM studies, but would be useful for better understanding shopping patterns. Further, understanding such spatial dynamics could be strategic for choosing intervention locations that may be best suited to the needs of the population. Also, much higher traffic was observed for Cadillac market (53 vs. 16 shoppers per day at Guybourg). To ensure economic viability of UPS in poor neighbourhoods while improving access to healthy food, it seems worthwhile to locate these services near transit hubs. Such locations can serve a larger and more diverse clientele, thus reaching the double goal of increasing access to FVs for disadvantaged populations and of optimizing economic survival by also including consumers with higher purchasing power. Those observations concurred with Sadler’s findings emphasizing that location of a food market in a prominent and central location is key to increasing access by attracting both residents and broader clientele [35].

Besides accessibility variables, the only other variable that we found to be related to purchase indicators was place of birth, consistent with Canadian data highlighting that immigrants may have healthier eating habits [102,103]. Participants born in Canada bought fewer FV portions, less variety and spent less per purchase. Reflecting the composition of the adjacent neighbourhoods, most shoppers were born in Canada (>75% in our study versus 77% in 2016 neighbourhood statistics) [104]. The proportion of shoppers living on a low-income was also similar to that of the adjacent neighbourhoods (23.0% vs. 16.6% to 20.3%). This suggests that the interventions reached a variety of socioeconomic groups. However, most shoppers were highly educated. This finding is distinct from observations made in low-income communities by Jennings and colleagues in the UK [34], but coincides with those made by Woodruff in the UK, USA and Australia [39].

In our study, the Square application database allowed us to access data for each food item rather than only by food category, unlike existing databases that use barcodes (e.g., Nielsen). Furthermore, we connected purchase data to both point of sale and to the individual shopper. This made it possible to look at three distinct FV purchase indicators: the number of FV portions, the variety of FV purchased per transaction, and the amount spent. The number of FV portions and amount spent are indicators initially developed by the Institut National de Santé Publique du Québec to assess and describe the quality of food consumption in Quebec [57]. These indicators are used in other settings and countries. We encourage future studies to adopt these standards to facilitate comparison between studies, and possibly even compare results to data obtained from existing databases using barcodes such as Nielsen. However, it is still possible to extract raw quantities for broader comparisons. Several countries are implementing laws requiring small companies to adopt electronic billing systems [76], which will facilitate this type of study.

The organization operating the market reported a very positive experience with the Square application as it was an easy-to-use technology. The application also enabled the analysis of traffic and peak hours. Other similar initiatives in Montréal also showed interest in the system for future interventions. However, sellers need to pay user fees, which could be an obstacle for usage in smaller settings. For the customer, the application does not add constraints compared to classical retailers. Furthermore, shoppers could pay either cash or by credit card, which is not always the case in that type of venue, thus increasing economic accessibility.

The use of an electronic payment application also enables standardization in purchase data entry. In addition, manual recording is not only time-consuming and resource-intensive but also presents the risk of missed transactions, especially during busy hours [43,56,59,71,72,75,105]. Alia et al. highlighted the potential for electronic application to collect purchase data in UPS [56].

Like every new method, the use of Square also presented a few challenges. We found that data accuracy depends largely on the research assistants, who need to be properly trained to avoid mistakes when documenting details of the transaction (date, time, and list of some items). In our case, the FV stands studied did not have very high traffic, facilitating documentation and the subsequent retrieval of the right transaction. However, in venues with higher traffic, we recommend the procedure to be adapted to ensure survey data can accurately be linked to transaction data. Another possible source of error might be linked to the accuracy and precision of the vendor who registers the purchases into the application. However, as the application also helps to identify the right amount to be paid, there is no reason the vendor would purposefully generate errors.

We would like to highlight several strengths of this study. This is one of the first studies that attempts to automatically quantify purchasing data on a local and provisional food source, on two separate sites, taking into account several spatial variables. The few studies that examined sales data from farmers’ market interventions mainly focused on the impact of financial incentives for purchasing FV [43,59,61,62,63,64,65,66]. These reported the amount spent by shoppers and/or type of purchase, but did not assess other factors associated with FV purchase. One limitation is that it is difficult to make comparisons between studies since prices and types of FV can vary widely between countries and regions. The proposed purchase analysis allows a gap in the literature to be filled as it enables the analysis of quantities of FV purchased, which helps understanding dietary health practices and enables comparison across contexts. Furthermore, the available information makes the calculation of the average price per portion of FV possible. Another strength of the method is the consideration of the edible portion of FV rather than the total weight purchased, which is more relevant in terms of consumption.

Several limitations need to be underlined. The sample size was relatively small and models may have lacked statistical power. Even though similar studies use the same type of cross-sectional design, results need to be interpreted with caution as we cannot establish any causal relationships. As this study occurred in a natural setting, several parameters could not be controlled by the researchers, thus limiting deeper analyses. While Guybourg shoppers reported a higher visit frequency to the market (one-time-only visits were reported as 62.7% in Cadillac vs. 42.9% in Guybourg), interpretation is limited as implementation duration was different during the 2016 season (14 days at Cadillac vs. 45 days at Guybourg). Awareness of the market and adoption among the local population also differed (data not shown). Although Cadillac opened later in the season (Sept–Oct), data at both markets were collected during the same period. An overview of the data confirmed that available species of FV were globally the same for both markets during the study period. Furthermore, the use of the Square application could present acceptability issues for some merchants accustomed to accepting only cash because recording every transaction may have repercussions for tax returns. This could be a factor limiting its use and expansion in intervention studies or public health interventions.

## 5. Conclusions

This study presents an innovative method for measuring FV practices using purchase data from a mobile application that allowed us to analyse the links between socioeconomic factors, subjective and objective variables of food environments, spatial- and mobility-related factors, and purchase of FV among participants in disadvantaged neighbourhoods. Because it does not require an optical scan system, this purchase analysis method could be easy to apply in both unconventional points of sales research and in public health interventions. In order to evaluate more accurately the impact on diet of food environment interventions, we need to deepen our understanding of the complexity of food shopping practices. With many quasi-natural experimental opportunities arising from local food environment interventions and increasing adoption of UPS, the use of objective indicators of food shopping practices as proposed here offers new ways to explore food-related health and inequities. The tools and indicators used in this study could be easily applied to other countries and other settings.

## Figures and Tables

**Table 1 ijerph-17-00088-t001:** Descriptive analyses of participants recruited in the intervention markets (*n* = 257).

Variables	Total Sample (*n* = 257)
**Sociodemographic characteristics**	
**Age, years** (%)	
18–44	47.9
45–64	36.6
65 and over	15.6
**Sex** (%)	
Male	77.8
Female	21.8
Missing	0.39
**Household size (mean, SD)**	2.2 (1.2)
**Household categories (person/household)**	
1	34.6
2	32.7
3	15.6
More than 3	17.1
**Education** (%)	
High school or less	24.1
Trade school or pre-university college	37.4
University	38.5
**Household under the LIM** (%)	
Yes	23.0
No	61.9
Missing	15.2
**Born in Canada** (%)	
Yes	78.2
No	21.8
**Spatial and mobility-related variables**	
**Car access** (%)	
Yes	55.6
No	44.4
**Home-to-market distance, meters**	1992.9 (3590.5)
**Residing ≤ 1 km from the market (%)**	
Yes	66.2
**Unhealthy/healthy densities ratio (mean, SD)**	2.8 (0.6)
**Market on usual travel route** (%)	
Yes	77.8
No	21.4
Missing	0.8
**FV consumption, perceived access and shopping habits**	
**FV consumption per day** (mean, SD) (0.8% missing values)	4.6 (2.1)
**Eating at least 5 FV per day, %**	
Yes	41.6
**Easy to find fresh FV of good quality in own neighbourhood (%)**	
Agree	47.1
More or less agree	15.2
Disagree	37.0
Missing	0.8
**Fresh FV are not affordable in own neighbourhood (%)**	
Agree	28.4
More or less agree	14.8
Disagree	52.9
Missing	3.9
**Market shopping habits (%)**	
Less than once a month	22.6
1 to 3 times/month	25.6
Once a week or more	49.8
Missing	2.0
**Number of visits at the intervention market (%)**	
One visit	58.4
Between one visit total and one visit per month	29.6
More than one visit per month	12.06
**Purchasing practice analysis**	
**Number of FV portions purchased, mean (SD)**	11.7 (13.8)
**Variety of FV purchased, mean (SD)**	2.8 (2.0)
**Amount spent (CAD),**	
**mean (SD)**	5.0 (3.9)
**Median**	4.1

NB: No missing values when the Missing category is not mentioned. LIM: Low-income measure; FV: fruits and vegetables/SD: Standard Deviation.

**Table 2 ijerph-17-00088-t002:** Results for multivariate linear regression modeling of number of FV portions purchased (*n* = 257).

Variables	Estimate (CI 95%)	*p*-Value
**Market location**		0.003
Cadillac	Ref	
Guybourg	0.87 (0.30; 1.43) **	
**Age**		0.39
18–44	Ref	
45–64	−0.37 (−0.85; 0.11)	
65 and over	−0.09 (−0.77; 0.58)	
**Sex**		0.66
Male	Ref	
Female	0.12 (−0.41; 0.65)	
**Household size**	−0.12 (−0.34; 0.09)	0.26
**Education**		0.71
High school or less	−0.03 (−0.62; 0.56)	
Trade school or pre-university college	0.15 (−0.35; 0.64)	
University	Ref	
**Household under the LIM**		0.82
No	Ref	
Yes	−0.06 (−0.62; 0.56)	
**Born in Canada**		0.004
No	Ref	
Yes	−0.81 (−1.36; −0.25) **	
**Car access**		0.89
Yes	Ref	
No	0.03 (−0.46; 0.53)	
**Home-to-market distance, 100m**	−0.004 (−0.01; 0.002)	0.18
**Unhealthy/healthy densities ratio**	−0.01 (−0.39; 0.36)	0.94
**Market on usual travel route**		0.03
No	Ref	
Yes	−0.58 (−1.10; −0.07) *	
No	Ref	
**Fruit and vegetable consumption**	0.03 (−0.08; 0.13)	0.60
**Market shopping habits**		0.80
Less than once a month	0.14 (−0.39; 0.68)	
1 to 3 times/month	0.004 (−0.48; 0.49)	
Once a week or more	Ref	
**Easy to find fresh FV of good**		0.10
**quality in own neighbourhood**		
Agree	Ref	
More or less agree	0.66 (0.06; 1.26) *	
Disagree	0.23 (−0.25; 0.71)	
**Fresh FV are not affordable in own neighbourhood**		0.18
Agree	−0.26 (−0.77; 0.26)	
More or less agree	−0.33 (−0.94; 0.28)	
Disagree	Ref	
**Number of visits at the intervention market**		0.22
One visit	−0.60 (−1.28; 0.08)	
Between one visit total and one visit per month	−0.53 (−1.25; 0.18)	
More than one visit per month	Ref	

LIM: Low-income measure; FV: fruits and vegetables.* *p* < 0.05; ** *p* < 0.01.

**Table 3 ijerph-17-00088-t003:** Results from multivariate linear regression modeling of amount spent (*n* = 257).

Variables	Estimate (CI 95%)	*p*-Value
**Market location**		0.99
Cadillac	Ref	
Guybourg	−0.01 (−1.35;1.32)	
**Age, years**		0.48
18–44	Ref	
45–64	−0.66 (−1.80;0.48)	
65 and over	−1.15 (−2.76; 0.46)	
**Sex**		0.99
Male	Ref	
Female	0.01 (−1.24; 1.25)	
**Household size**	−0.26 (−0.77; 0.24)	0.31
**Education**		0.92
High school or less	0.25 (−1.17; 1.68)	
Trade school or pre-university college	0.37 (−0.80; 1.54)	
University	Ref	
**Household under the LIM**		0.37
No	Ref	
Yes	−0.57 (−1.85; 0.70)	
**Born in Canada**		0.04
No	Ref	
Yes	−1.39 (−2.69; −0.08) *	
**Car access**		0.58
Yes	Ref	
No	−0.33 (−1.50; 0.85)	
**Home-to-market distance, 100m**	−0.01 (−0.02; 0.01)	0.22
**Unhealthy/healthy densities ratio**	−0.32 (−1.19; 0.55)	0.47
**Market on usual travel route**		0.11
No	Ref	
Yes	−0.99 (−2.21; 0.22)	
**Fruit and vegetable consumption**	0.17 (−0.08; 0.42)	0.18
**Market shopping habits**		0.18
Less than once a month	0.12 (−1.13; 1.38)	
1 to 3 times/month	−0.77 (−1.94; 0.39)	
Once a week or more	Ref	
**Easy to purchase fresh FV on usual travel route**		0.16
Agree	Ref	
More or less agree	−2.21 (−4.85; 0.42)	
Disagree	0.39 (−0.70; 1.49)	
**Fresh FV are not affordable in own neighbourhood**		0.20
Agree	−0.40 (−1.62; 0.83)	
More or less agree	−1.07 (−2.49; 0.35)	
Disagree	Ref	
**Number of visits at the intervention market**		0.01
One visit	−2.41 (−4.02; −0.81) **	
Between one visit total and one visit per month	−1.78 (−3.47; −0.10) *	
More than one visit per month	Ref	

LIM: Low-income measure; FV: fruits and vegetables. * *p* < 0.05; ** *p* < 0.01.

**Table 4 ijerph-17-00088-t004:** Results of Poisson regression modeling of variety of FV purchased (*n* = 257).

Variables	Estimate (CI 95%)	*p*-Value
**Market location**		0.49
Cadillac	Ref	
Guybourg	0.07 (−0.13; 0.28)	
**Age**		
18–44	Ref	
45–64	−0.10 (−0.28; 0.08)	0.28
65 and over	−0.13 (−0.39; 0.12)	0.30
**Sex**		0.47
Male	Ref	
Female	0.07 (−0.13; 0.28)	
**Household size**	−0.10 (−0.17; −0.01) *	0.02
**Education**		
High school or less	0.05 (−0,18; 0,28)	0.68
Trade school or pre-university college	0.15 (−0,04; 0,33)	0.12
University	Ref	
**Household under the LIM**		0.35
No	Ref	
Yes	−0.10 (−0.32; 0.11)	
**Born in Canada**		0.02
No	Ref	
Yes	−0.24 (−0.44; −0.03) *	
**Car access**		0.39
Yes	Ref	
No	−0.08 (−0.27; 0.11)	
**Home-to-market distance, 100m**	−0.001 (−0.004; 0.001)	0.29
**Unhealthy/healthy densities ratio**	0.04 (−0.11; 0.19)	0.59
**Market on usual travel route**		<0.01
No	Ref	
Yes	-0.25 (-0.43; -0.07) *	
**Fruit and vegetable consumption**	0.04 (0.001; 0.08) *	0.04
**Market shopping habits**		0.39
Less than once a month	0.09 (−0.11; 0.29)	0.63
1 to 3 times/month	−0.05 (−0.25; 0.15)	
Once a week or more	Ref	
**Easy to purchase fresh FV on usual travel route**		
Agree	Ref	
More or less agree	−0.70 (−1.28; −0.12) *	0.02
Disagree	0.03 (−0.14; 0.20)	0.72
**Fresh FV are not affordable in own neighbourhood**		
Agree	−0.12 (−0.31; 0.08)	0.23
More or less agree	−0.27 (−0.52; −0.02) *	0.04
Disagree	Ref	
**Number of visits at the intervention market**		0.004
One visit	−0.34 (−0.57; −0.11) **	0.009
Between one visit total and one visit per month	−0.33 (−0.58; −0.08) **	
More than one visit per month	Ref	

LIM: Low-income measure; FV: fruits and vegetables. * *p* < 0.05; ** *p* < 0.01.

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
