# Peer review of "Fruit and Vegetable Purchases in Farmer’s Market Stands: Analysing Survey and Sales Data"

_ijerph, 2019, doi:10.3390/ijerph17010088_

Round 1

Reviewer 1 Report

The paper contains new information about the purchasing practices at the Cadillac and Guybourg market in Montreal. A regional study like this could provide a useful additional case study to extend the literature on consumer purchasing practices. However, there are a number of weaknesses in the current article and it is not suitable for publication in its current form.  The paper suffers from multiple problems that prevent me from recommending its publication.

The most confusing is the aim of the study. In the abstract it is developing of an innovative method for measuring FV purchasing practices and in the end of the introduction section – how accessibility measurements, individual perception on food access and other mobility-related factors are associated with FV purchasing practices or the elaborating three purchase indicators. I would like to know why is so important to elaborate novel indicators. What is the aim to use the novel tolls and the indicators in Canada and in other countries? What are the benefits and practical implications?  

I suppose, it is a methodologic paper with the validation of the presented method. But it was not clearly indicated.

The current description of the results must be rewritten because the results are not presented clearly. I appreciate the statistical analysis and I see potential scientific value of the paper. However, the result description is not sufficient in the present form.

The manuscript provides a snapshot of purchasing practices in Montreal. I recommend to present the results in a broader perspective. If you did it your paper would be more valuable for readers. In addition, the conclusions do not tie together the aim of the study and revealed findings.

I did not recognize any possibility of using the research in practice. Please try to consider it.

Please ensure the positive impact of your findings and experience on society in Montreal and other places.

Reviewer 2 Report

Interesting piece of research and well structured article. Minor improvements needed-

Introduction needs improvement with regard to including research specific to Canada, for example the current state of farm markets etc.  

Elaborate questionnaire design, were any changes made to the questionnaire to suit the target population? was it pilot tested for the population?

What is the reliability coefficient of the survey?

Round 2

Reviewer 1 Report

Dear Authors

Well done job. I am pleased to see such amended paper.

This manuscript is a resubmission of an earlier submission. The following is a list of the peer review reports and author responses from that submission.

Round 1

Reviewer 1 Report

I am unclear on the aims of this paper and the major theoretical contributions it is trying to make.

First, with respect to the aims:

·         The abstract says the aim is compare shoppers’ profiles and factors associated with purchasing

·         The end of the intro leads with an aim of presenting a novel method to assess FV purchasing, and states that beyond presenting this method, the study also aims to compare shoppers’ profiles and factors associated with purchasing

·         The aims in the discussion are in line with those stated in the abstract

·         It is not until line 283-284 in the discussion that we learn that part of these aims was to specifically examine one FM located on a transit route, and another that is in a food desert.

The confusion around the aims left me unclear whether this was intended to be a methodologic paper and I could not gain clarity around this by reading the paper.  The final conclusion of the abstract relates to methods, and the intro sets up a methodologic gap, however the methods section does not describe the novel method in depth and the discussion does not mention the novel method until the section on strengths and limitations.  I was very interested in the method, but could not find details such as how the authors negotiated access to consumer purchasing data via the Square app (beyond a mention of a partnership with YQQ), whether all shoppers were required to have their purchases recorded by the Square app as a condition of shopping at the FM and how that was negotiated, and more details on the data contained within the app, how common use of the app is and its feasibility for use in other studies in other settings, nations, etc. It took me several reads to realize that it was the vendors and not the shoppers who used the Square app. What are the potential implications, if any, of the use of the app in terms of the types of vendors or shoppers who would use these FM? Would some shoppers be hesitant to shop there knowing data would be recorded by the app or might the app be a barrier for some vendors who are less technologically savvy?

Intro:

·         The beginning of the introduction appears to conflate individual and neighbourhood-level disadvantage.  The first few sentences relate to individually disadvantaged groups, but there is a sudden leap to disadvantaged neighbourhoods. 

·         There are also references to “evaluating the effectiveness of FM”, but it is not clear what aspects of FM need to be evaluated (i.e. their introduction in low SES neighbourhoods…?) and effectiveness for what (i.e. increasing FV intake?).

·         Line 70 discusses a gap that this study does not actually fill since this study does not examine the effectiveness of an intervention

Methods:

·         Line 103-105 are very confusing to the reader as the reader has not been told that this study is about an intervention, nor does this study test the impact of any intervention.  What is the nature of this intervention?

·         Lines 111-119: This is a key issue of concern as one FM was only open Sept-Oct, which is the end of the season and the types and amounts of produce are likely to be substantially limited, i.e. likely only root vegetables available.  The other FM was open July – Oct and would have had a much greater variety of produce available.  It is not valid to compare purchases across two very different selling periods in 2 different FM.

Discussion:

·         Line 323-324: if these data are available on types of FV purchased, then I don’t understand why the authors are merely speculating as to types rather than presenting these data.  This is something that could significantly enhance the paper and has potential to present some novel insights.

As currently written this paper cannot stand on its own as a methodologic paper, however neither does it present substantive new data related to FM purchasing that represent a novel contribution to the literature. The paper merely presents data to show that some of characteristics of FM shoppers do and do not differ in 2 different FM, which is something I would have expected. Similarly, some of the factors that influence purchasing in these 2 FM differ and some do not.  I am not clear on the novel contribution here. Moreover, the issue of comparing sales in 2 different FM during 2 different periods of the produce season poses a substantial concern.

Reviewer 2 Report

The results are not presented clearly. The current description of the results must be rewritten. It is insufficient in the present form.

I think that discussion section of the manuscript should be enriched with more references. The authors stated that the study was the first in Montreal. How about other countries? Please add information about it in selected countiries.

 The conclusions do not tie together the elements of the paper.

The conclusions should be presented in a broader context. The authors made a huge effort to evaluate such a big group of shoppers, however, I cannot see any scientific value in it. The results are a little bit predictable. The manuscript provides a snapshot of buying fruits and vegetable in specific area. I recommend to present the results in a broader, international perspective.

The conclusion section should be rewritten. There is no explanation of the importance or relevance of the study.  

 Comments:

 Some detailed suggestions include: The paper contains a lot of typing flaws e.g. p. 11, L 310 space between words („home[93]”) nedds to be added. Please correct them.

Reviewer 3 Report

General comment

Authors accurately demonstrated how physical access to the markets, perceptions of access to  FV, affects FV consumption and FV purchases in three specific neighbourhoods (Louis-Riel, Longue-Pointe and Guybourg). 

The manuscript was well articulated and easy to read and comprehend. however, the material and method sections including the discussion could be further shortened.